# Alternate Special Stains for the Detection of Mycotic Organisms in Oral Cyto-Smears—A Histomorphometric Study

**DOI:** 10.3390/microorganisms10061226

**Published:** 2022-06-15

**Authors:** S. V. Sowmya, Dominic Augustine, Beena Hemanth, Arcot Gopal Prathab, Ahmed Alamoudi, Hammam Ahmed Bahammam, Sarah Ahmed Bahammam, Maha A. Bahammam, Vanishri C. Haragannavar, Sonia Prabhu, Shankargouda Patil

**Affiliations:** 1Department of Oral Pathology & Microbiology, Faculty of Dental Sciences, M.S. Ramaiah University of Applied Sciences, Bangalore 560054, Karnataka, India; drsowmya25@gmail.com (S.V.S.); dominic2germain@gmail.com (D.A.); vani.haragannavar@gmail.com (V.C.H.); ssp046.ds.ds17@msruas.ac.in (S.P.); 2Department of Microbiology, Ramaiah Medical College, MSRIT Post, Bangalore 560054, Karnataka, India; drbeenahemanth@yahoo.co.in (B.H.); dragprathab@yahoo.co.in (A.G.P.); 3Oral Biology Department, Faculty of Dentistry, King Abdulaziz University, Jeddah 80209, Saudi Arabia; ahmalamoudi@kau.edu.sa; 4Department of Pediatric Dentistry, Faculty of Dentistry, King Abdulaziz University, Jeddah 80209, Saudi Arabia; habahammam@kau.edu.sa; 5Department of Pediatric Dentistry & Orthodontics, College of Dentistry, Taibah University, Medina 42353, Saudi Arabia; sbahammam@taibahu.edu.sa; 6Department of Periodontology, Faculty of Dentistry, King Abdulaziz University, Jeddah 80209, Saudi Arabia; mbahammam@kau.edu.sa; 7Executive Presidency of Academic Affairs, Saudi Commission for Health Specialties, Riyadh 11614, Saudi Arabia; 8Department of Maxillofacial Surgery and Diagnostic Sciences, Division of Oral Pathology, College of Dentistry, Jazan University, Shwajra Campus, Jazan 45412, Saudi Arabia

**Keywords:** *Candida albicans*, *Aspergillus flavus*, *Rhizopus oryzae*, Safranin-O, Gomori’s Methenamine Silver, periodic acid–Schiff, Alcian Blue, morphometry, image analysis

## Abstract

In the wake of the COVID-19 pandemic, fungal infections of the maxillofacial region have become prevalent, making their accurate diagnosis vital. Histopathological staining remains a simple, cost-effective technique for differentiation and diagnosis of the causative fungal organisms. The present study aims to evaluate the staining efficacy of Periodic Acid-Schiff (PAS), Alcian Blue, Safranin-O and Gomori’s Methenamine Silver (GMS) on fungal smears. This research work also attempts to study the morphometric characteristics of *Candida albicans*, *Aspergillus flavus*, *Rhizopus oryzae*. *Candida albicans*, *Aspergillus flavus* and *Rhizopus oryzae*, 10 smears each, were stained using PAS, Alcian Blue, Safranin-O and GMS. The morphological characteristics and staining efficacy were examined, and semi-quantitative scoring was performed. *Candida albicans*, *Aspergillus flavus* and *Rhizopus oryzae* were stained for the first time with Safranin-O. The morphometric traits were then analyzed using an image analysis software. Safranin-O provided the most reliable staining efficacy amongst the stains and optimum morphological definition for all three organisms. Safranin-O was found to be superior to PAS and GMS, ensuring detection of even the most minute mycotic colonies. The hyphae of *Aspergillus flavus* to be the largest, and the spores and fruiting body of *Rhizopus oryzae* were found to be the largest amongst the three organisms compared. Early and accurate diagnosis of fungal infections can significantly reduce morbidity in orofacial fungal infections.

## 1. Introduction

Fungal infections of the maxillofacial region used to be an uncommon occurrence that have become commonplace since the start of the COVID-19 pandemic. This can be attributed to the prevalence of larger populations of high-risk patients, such as those with comorbidities and immunosuppressive states [1]. The causative organisms of maxillofacial fungal infections are usually Candida and Aspergillus species and, more recently, Mucorales. Superficial infections such as Candidiasis and Aspergillosis can be tackled conservatively as opposed to deep fungal infections such as mucormycosis, which can be difficult to diagnose and requires an aggressive approach for management [2].

These fungal infections can also present as dual infections where a combination of fungi is responsible. The diagnosis of these dual infections can be challenging for the practitioner, as an inappropriate treatment plan may lead to delayed intervention and the development of further complications. In cases with extensive tissue destruction and necrosis, fewer organisms may be present in the smear, and these organisms may be masked by the surrounding structures. In this situation, it is important that the organisms that are present can be identified and differentiated by the pathologist to arrive at an accurate diagnosis. Timely diagnosis and management are especially important in immunocompromised patients due to the higher rates of mortality and morbidity. COVID-19-associated mucormycosis was said to have a mortality rate of nearly 31–50% [3]. The post-COVID-19 era has shown a surge in mucormycotic infection of the orofacial region that causes significant bone destruction in patients, which are often fatal due to delayed diagnosis [4]. Most often, oral pathologists receive oral smears of potentially life-threatening fungal infections, and on many occasions the smears are masked by inflammatory cells, exfoliated epithelial cells and red blood cells, making the diagnosis challenging. The use of special stains is a simple, quick and cost-effective technique that can aid in the early diagnosis of superficial and deep fungal infections and minimize the extent of destruction, leading to more favorable outcomes and better prognoses for patients.

Conventionally used stains such as hematoxylin and eosin (H&E) exhibit poor staining efficacy in sections with very few organisms, making the diagnosis difficult [5]. To overcome this major drawback, the efficacy of an alternative stain such as Safranin-O on fungal organisms was explored. Alcian Blue is another alternative stain to be assessed alongside Safranin-O, and the staining efficacy can be compared to conventionally used stains for fungi, such as PAS and GMS. This can help identify the stain that provides the maximum staining efficacy and morphologic differentiation to aid in early and accurate diagnosis of the causative organism. Morphometric traits, such as the shape and size of the fungal organisms, can be important identifying features for the pathologist. The different morphologic characteristics of each organism and the range of presentation of the spores and hyphae were explored after histopathologic staining to further support the findings.

The current study aimed to evaluate the staining efficacy (staining intensity and morphologic differentiation) of Safranin-O, PAS, Alcian Blue and GMS on smears of *Candida albicans*, *Aspergillus flavus* and *Rhizopus oryzae* organisms. The present study also evaluated the morphological characteristics of *Candida albicans*, *Aspergillus flavus* and *Rhizopus oryzae* using these four stains through an image analysis software.

## 2. Materials and Methods

### 2.1. Sample Collection

The study was conducted by the department of Oral Pathology and Microbiology, Faculty of Dental Sciences, MS Ramaiah University of Applied Sciences, Bangalore. Fungal organisms, *Candida albicans*, *Aspergillus flavus* and *Rhizopus oryzae*, were cultured and 10 smears of each of the organisms were obtained from the Department of Microbiology, Ramaiah Medical College, Bangalore, and stained using PAS, Alcian Blue, Safranin-O and GMS.

### 2.2. PAS Staining

The smear was oxidized in 0.5% periodic acid solution for 5 min. The slide was then rinsed using distilled water and placed in Schiff reagent for 15 min. The smear was washed with lukewarm tap water for 5 min and counterstained with Mayer’s hematoxylin solution for 1 min; following this, the smear was washed under tap water for 5 min and left to dry [6].

### 2.3. GMS Staining

The smear was oxidized with 4% aqueous chromic acid at room temperature for 1 h and the smear was washed under water for a few seconds. The smear was then stained with 1% sodium metabisulphite for 1 min and washed under smoothly running tap water for 3 min and rinsed thoroughly in distilled water. The slide was then placed in silver solution in water bath at 60 °C for 15–20 min until the smear turned dark brown, and then was rinsed with distilled water. A 0.2% gold chloride solution was added to the smear and left for 2 min, after which the smear was rinsed well with distilled water. The smear was then treated with 2% sodium thiosulphate for 2 min and then washed under smoothly running tap water for 5 min. Light-green solution was then used to counterstain the smear for 15 s and then rinsed to remove the excess alcohol before allowing it to dry [7].

### 2.4. Safranin-O Staining

The slide was stained with Weigert’s iron haematoxylin working solution for 10 min and washed under running tap water for 10 min. The smear was then stained using fast green solution for 5 min and rinsed quickly with 1% acetic acid solution for about 10–15 s. A 0.1% Safranin-O solution was then used to stain the smear for 5 min and ethyl alcohol, absolute ethyl alcohol and xylene were used to clear and dehydrate the smear by using 2 changes each for 2 min each [8,9,10,11,12].

### 2.5. Alcian Blue Staining

The slide was stained with Alcian Blue solution for 15 min and washed well under running tap water for 15 min. The slide was then rinsed in distilled water and counterstained using neutral red stain for 1 min. Absolute alcohol was then used to dehydrate the smear [13].

### 2.6. Analysis of Staining Efficacy in Fungal Organisms

The fungal smeared slides were examined at a magnification of ×100 and ×400 using an Olympus Optical Research Microscope (BX53F2, Tokyo, Japan) with a Jenoptik Progres Gryphax Arktur USB 3.0 microscope camera (Jena, Germany) to determine the morphological characteristics and staining efficacy with each of the special stains. The staining efficacy was measured using two parameters—the mean staining intensity and the morphologic differentiation provided by each of the stains. The evaluation of staining efficacy was carried out by two independent pathologists whose slides were blinded during interpretation and the mean values were considered as the final scores. The mean staining intensity was measured by scoring each parameter, namely the spores, hyphae and yeast/conidia/sporangia as follows: 0—not stained; 1—mildly stained; 2—moderately stained; 3—intensely stained. For the morphologic differentiation, the size of the spores, the shape of the spores, the nature of the hyphae (pseudohyphae/septate/non-septate) and their branching (angulation/germ tube formation) were scored as follows: 1—difficult to appreciate; 2—moderately differentiable; 3—easily differentiable. A semi-quantitative score was obtained for the staining efficacy. The values were tabulated and assessed for statistical significance using SPSS software version 20, IBM, New York.

### 2.7. Morphometric Analysis of Fungal Organisms

The morphometric traits, such as the shape and the size of the spores, hyphae and fruiting bodies, were analyzed using the Jenoptik Gryphax image analysis software v2.1, Jena, Germany, via line-measurement tool/free-form tool and a 3-point-circle measurement tool. The diameters of the smallest and largest spores were measured for each of the organisms and similarly the width of the smallest and largest hyphae was measured. A 3-point-circle measurement tool was used to measure the diameter of the spores of *Candida albicans*. The line-measurement tool was used to measure across the greatest length of the spores of *Aspergillus flavus* and *Rhizopus oryzae*. The line-measurement tool was also used to draw a line across the hyphal structure to measure the width of the hyphae. The surface area of the smallest and largest fruiting bodies of *Aspergillus flavus* were measured using the 3-point-circle measurement tool and the diameter of the smallest and largest fruiting bodies of *Rhizopus oryzae* were measured using the free-form tool. The morphometry was performed on 50 high-power fields and the average values obtained were tabulated. These values were used to arrive at a size range for each of the parameters for all three organisms.

## 3. Results

### 3.1. Comparison of Mean Staining Intensity

Safranin-O exhibited the most intense staining with respect to the spores, hyphae and fruiting bodies of all three organisms (*p* = 0.001). (Table 1, Table 2 and Table 3) GMS was comparable to Safranin-O for *Aspergillus flavus*. Alcian Blue was comparable to PAS for *Candida albicans* and *Aspergillus flavus*. Alcian Blue provided comparable staining intensity to that of Safranin-O for fruiting bodies of *Rhizopus oryzae*.

### 3.2. Comparison of Mean Morphologic Differentiation

#### 3.2.1. Spore Size

Morphologic differentiation was easily appreciated to assess the spore size of *Candida albicans* using Safranin-O and PAS staining. The organisms appeared moderately differentiated with Alcian blue but were difficult to appreciate in GMS-stained slides. *Aspergillus flavus* was easily appreciated in all 10 smears using Safranin-O, PAS and GMS, whereas it was moderately differentiable in Alcian blue staining. *Rhizopus oryzae* was easily appreciated in all the 10 smears using Safranin-O, PAS and GMS, and it was moderately differentiable in Alcian Blue staining. There was a statistically significant difference obtained (*p* = 0.001). (Table 4)

#### 3.2.2. Spore Shape

There was superior morphological differentiation observed in *Candida albicans* with Safranin-O and PAS staining as compared with Alcian Blue, which was moderately differentiated, and GMS, where the spore shape was difficult to appreciate. For *Aspergillus flavus* and *Rhizopus oryzae*, morphological differentiation was appreciated well using Safranin-O and PAS, it was moderate in differentiation using Alcian Blue, and could not be well appreciated using GMS staining. There was a significant difference in the morphological differentiation of the spore shape observed between the four stains in both these mycotic organisms (*p* = 0.001) (Table 5).

#### 3.2.3. Nature of Hyphae

It was observed that the morphological differentiation of the non-septate *Candida albicans* hyphae was easily differentiable in Safranin-O, was moderately appreciated in PAS and difficult to appreciate in Alcian Blue and GMS, this difference was statistically significant (*p* = 0.001). The septate hyphae of *Aspergillus flavus* were easily appreciated in Safranin-O staining, moderate in Alcian Blue and PAS and not well appreciable in GMS. Safranin-O and PAS staining showed easily appreciable hyphae of *Rhizopus oryzae*, whereas it was difficult to appreciate in Alcian Blue and GMS staining (Table 6).

#### 3.2.4. Nature of Branching

Germ tube formation was easily appreciated using Safranin-O, but moderately differentiated using PAS and difficult to appreciate using Alcian Blue and GMS staining. The branching of *Aspergillus flavus* was easily appreciated using Safranin-O, moderately appreciated in Alcian Blue and PAS and poorly appreciated in GMS staining. The morphological differentiation of branching of *Rhizopus oryzae* was easily appreciated using Safranin-O and PAS staining. There was difficulty in appreciating the branching using Alcian Blue and GMS staining. There was a significant difference in the morphological differentiation between the different stains (*p* = 0.001) (Table 7).

### 3.3. Comparison of Staining Efficacy

Safranin-O provided the most reliable staining intensity and overall morphological definition for all three organisms. PAS and Alcian Blue were comparable to each other in terms of staining efficacy for *Candida albicans* and *Aspergillus flavus*. PAS stained *Rhizopus oryzae* more effectively than Alcian Blue and GMS. The staining efficacy of GMS was comparable to that of Safranin-O for *Aspergillus flavus* (Figure 1).

### 3.4. Comparison of Morphometric Traits

Safranin-O-stained images were considered for the morphometric analysis. The spores of *Aspergillus flavus* range from 9.71 μm to 15.96 μm in diameter and were ovoid in shape. The hyphae ranged from 8.60 μm to 25.85 μm in width and were septate, showing branching at acute angles. The fruiting bodies had surface areas ranging from 18.94 μm to 33.46 μm. The spores of *Candida albicans* were round in shape and ranged in diameter from 8.77 μm to 17.86 μm. The pseudohyphae were non-septate and ranged in width from 3.80 μm to 17.75 μm. The spores of *Rhizopus oryzae* were angular–ovoid in shape with a diameter ranging from 15.96 μm to 28.25 μm. The hyphae were non-septate and branching at right angles, with widths ranging from 2.43 μm to 17.16 μm. The fruiting bodies of Rhizopus were the largest, ranging in diameter from 98.67 μm to 124.80 μm, respectively. (Figure 2) The differences in size between the organisms were found to be statistically significant (Table 8).

## 4. Discussion

Superficial fungal infections and, more recently, due to the COVID-19 pandemic, deep fungal infections of the oral cavity have become a common occurrence [14]. Fungal infections of the head and neck region commonly include candidiasis, aspergillosis and mucormycosis. Mucormycosis is a life-threatening, invasive infection that is seen in patients with immunocompromised states and patients with comorbidities such as diabetes mellitus. Mucormycosis can be associated with concurrent candidiasis or aspergillosis, complicating the clinical presentation of the disease. The treatment modalities for deep infections such as mucormycosis are far more invasive and necessitate aggressive management and immediate intervention [15]. Failure to identify the organisms present in a fungal smear would result in an erroneous diagnosis with a heavy price to pay.

The prevalence of COVID-19 associated mucormycosis (CAM) was estimated to be 140 cases per million population, a massive burden that was calculated to be 80 times higher than the prevalence in developed countries. CAM was also said to constitute 0.3% of all COVID-19-associated co-infections [16,17,18]. The alarming rise in the number of cases necessitated the search for quicker and more effective means of diagnosis and management without compromising the quality of care that the patient receives.

While mucormycotic infections present with certain identifiable features, the presentation does not remain specific for the disease. Infection by fungi such as Aspergillus or Fusarium can present with identical clinical features, making the diagnosis puzzling [19,20]. The management protocol for other fungal infections cannot minimize or prevent the extensive tissue destruction that Mucormycosis brings, and hence the misdiagnosis of this infection can significantly increase the mortality and morbidity. Mistaking a fungal osteomyelitis for a bacterial osteomyelitis can lead to a delay in the administration of mainstay pharmacological agents such as amphotericin B, which can further impair the treatment outcome and survival rates of the patients.

Although technology has evolved enough to make diagnosis easier for the clinician, the question of affordability remains unanswered. In a developing country such as India, sophisticated diagnostic techniques such as PCR and immunofluorescence are not economically feasible for most of the population and hence, histopathological staining remains the gold standard for diagnosis. Special stains are an affordable alternative to the newer advancements and can be easily used in different settings [21].

In a dental setting, the use of special stains can be essential to help the clinician establish the appropriate treatment plan for each patient. Conventional stains such as H&E are not specific for fungi as the surrounding structures may also stand out or obscure the fungal elements. The staining efficacy of conventional stains is poor compared with that of special stains with an affinity for fungal elements, and hence the efficacy of novel, alternative special stains for fungi was explored. Safranin-O was taken into consideration to replace these conventional stains as it has a binding affinity for the glycoproteins present in the wall of the fungal organism, provides good staining efficacy and clearly highlights the morphology of the fungal organism against the surrounding structures. The use of Safranin-O as a special stain can aid in timely diagnosis and immediate administration of antifungal therapy and surgical debridement. This prevents dissemination of the infection and greatly improves the surgical outcome.

In routine practice, a fungal culture is the most preferred diagnostic method for fungal infections, but it is not without its drawbacks. Isolation of the fungi is difficult, especially in surface lesions due to the presence of both the primary pathogen as well as organisms that have secondarily invaded the lesion. Fungal cultures require a longer time period to allow the growth of the fungi and can delay the diagnosis. Furthermore, this method is more expensive and requires a good amount of expertise in the field to accurately culture and identify the organism [22,23,24]. Compared with these conventional methods, the use of smears to confirm the diagnosis is simple, cost-effective and does not require a sophisticated lab set up that is free from contamination. Therefore, this study explored the use of special stains to identify and assess the fungal organisms.

This study was carried out on fungal smears of three organisms, namely, *Candida albicans*, *Aspergillus flavus* and *Rhizopus oryzae*. The smears were stained using PAS, Alcian Blue, GMS and Safranin-O, which are special stains with a binding affinity for the fungal wall. The mean staining intensity and morphologic differentiation were observed under each stain to determine the staining efficacy.

In a study by Maiorano et al., the diagnosis of combined mucormycosis and aspergillosis was made by staining the tissue specimen with H&E, Giemsa, GMS and PAS stains. Microscopic examination showed large, branching, non-septate hyphae of *Rhizopus oryzae* stained dark brown–black with GMS, umbrella-like sporangiophores of Rhizopus stained light pink with Lugol’s iodine, and flower-like conidiophores of *Aspergillus flavus* stained ochre–light brown with Lugol’s iodine. The mycotic hyphae were difficult to distinguish from each other under GMS and the final diagnosis was made after noting the presence of the fruiting bodies of both the organisms [25]. A study by Smith et al. used plastic-embedded tissue sections that were stained with PAS and GMS. It was found that GMS-stained *Candida albicans* unevenly while PAS showed greater staining intensity as well as even staining of the organism. PAS was found to be superior to GMS for *Candida albicans* [26]. In another study by Padhi et al., tissue sections of mycetomas were stained using PAS, GMS, Brown and Brenn modification of Gram stain and Masson Fontana (MF) stain. Hyphae of *Aspergillus flavus* were found to be positively stained by GMS but did not take up the other stains well [27]. The diagnosis of a rare mycetoma caused by *Aspergillus flavus* was made after microscopically examining and identifying the organism.

Our study explored the use of Safranin-O, a special stain that is yet to be explored in the identification of fungal organisms. Compared with the other studies where conventional stains were found to be the most effective for identification, our study comprehensively compared four special stains, two of which are conventionally used for the diagnosis of fungal infections and two that were explored for the first time. Another point that sets this study apart is the finding that Safranin-O, a novel stain for fungi, provided the greatest staining efficacy for all three fungal organisms over conventional stains such as PAS and GMS. While some organisms in the other studies do not take up stains such as GMS well, all three organisms in our study were intensely and evenly stained by Safranin-O.

Microscopic examination of each of the organisms under the four special stains revealed that Safranin-O provided the greatest staining efficacy for all three organisms. The staining intensity of Safranin-O was superior to that of PAS, GMS and Alcian Blue for *Rhizopus oryzae*, making it the preferred stain for the diagnosis of Mucormycosis.

Fungal morphology plays an important role in classification and identification of the organisms. Morphometric analysis of the fungal organisms involves a study of the size and the shape simultaneously, providing a better understanding of the structure of the fungi and easier identification than by structure alone [28]. Morphometric evaluation using digital software allows the clinician to easily measure the organisms and compare the findings to the standard ranges, thereby making the diagnostic procedure simple. Such studies have been seldom performed.

The morphometric traits such as shape and size were observed using the Jenoptik Gryphax image analysis software and various tools such as the line-measurement tool, free-form tool and 3-point-circle measurement tool were used to calculate the morphometry to determine the key identifying features for each organism.

In a study by Nielsen et al., the effect of protein treatment on hyphal branching of *Aspergillus flavus* was determined. A conidium with multiple germ tubes or multiple hyphal tips on a single germ tube satisfied the criteria of branching. The largest square area covered by hyphal growth was considered and the number of branched and unbranched hyphae were counted [29]. In a study by Raas et al., biofilms of *Candida albicans* and Candida glabrata were photographed under a scanning electron microscope (SEM). The scanned images (×40–×100) were analyzed using FIJI software to determine the cell area and roundness [30]. A value of 0.1 for roundness indicated a perfect circle and values greater than 1.0 indicated oblong cells. Cell roundness analysis revealed that evidence of budding and pronounced roundness was observed more frequently in Candida glabrata as compared with *Candida albicans*, which exhibited a slightly more oblong shape [31]. In a study by Klich et al., cultures of *Aspergillus flavus* were examined under ×100 magnification to determine the diameter of the conidia, maximum conidiophore length and maximum vesicle diameter. The diameter of the conidia was found to be between 3.0 and 7.0 μm. The maximum conidiophore length was found to be between 340 and 1650 μm. The maximum vesicle diameter was found to be between 22 and 48 μm [32].

In our study, *Candida albicans*, *Aspergillus flavus* and *Rhizopus oryzae* were examined under ×100 magnification to identify morphologic traits that could be used to unmistakably diagnose the organism responsible for the infection. The current study reported the hyphae of *Aspergillus flavus* to be the largest, and the spores and fruiting body of *Rhizopus oryzae* were found to be the largest amongst the three organisms compared. Unlike other studies that studied singular parts of the organism, this study aimed to analyze the size, shape and structure of the spores, fruiting bodies and the hyphal elements. Values were calculated using Jenoptik Gryphax image analysis software and these values were used to arrive at a size range for each organism. This range can be used to further differentiate organisms with hyphae or spores that appear indistinguishable.

The three fungal organisms, namely *Candida albicans*, *Aspergillus flavus* and *Rhizopus oryzae*, were stained for the first time with Safranin-O. The fungal organisms were comprehensively stained with four different histochemical stains to eliminate any bias. Morphometry was performed to make this study comprehensive. We propose that Safranin-O should be used as a histopathological stain in oral fungal infections for easier identification of the organisms. Safranin-O is a cost-effective alternative to the conventionally used stains such as GMS. As a future scope, the results of the study can be tested by staining tissue sections taken from patients suspected to be infected with Mucormycosis using Safranin-O. The oral pathologist can employ Safranin-O in effective diagnosis of oral fungal infections, especially mucormycosis.

## 5. Conclusions

Safranin-O, a novel stain for fungi, was found to be superior to PAS, GMS and Alcian blue in staining *Candida albicans*, *Aspergillus flavus* and *Rhizopus oryzae*. The intense staining and enhanced morphologic definition ensured detection of even the most minute mycotic colony present. The superior staining efficacy of Safranin-O helps the pathologist differentiate the fungal organisms from each other as well as the surrounding tissues. The morphometric traits noted for each organism can be used to identify the causative organism accurately. Accurate identification of the organism and early intervention plays a vital role in medical and surgical management of the patients, reducing the amount of tissue destruction and minimizing the spread of infection, thereby improving their quality of life.

## Figures and Tables

**Figure 1 microorganisms-10-01226-f001:**
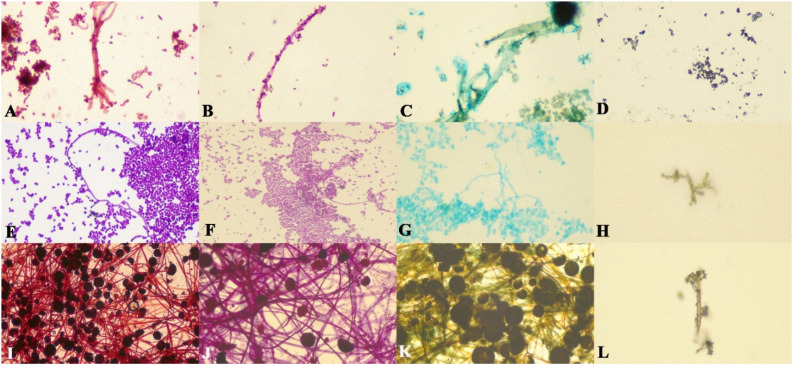
Photomicrographs of *Aspergillus flavus* showing: (**A**) branching hyphae and spores (×100 magnification, Safranin-O); (**B**) hyphae and spores (×100 magnification, PAS stain); (**C**) hyphae, spores and fruiting body (×100 magnification, Alcian Blue stain); (**D**) spores (×100 magnification, GMS stain). *Candida albicans* showing (**E**) spores, hyphae and germ tube formation (×100 magnification, Safranin-O); (**F**) hyphae and spores (×100 magnification, PAS stain); (**G**) hyphae and spores (×100 magnification, Alcian Blue stain); (**H**) germ tube formation (×100 magnification, GMS Stain). *Rhizopus Oryzae* showing (**I**) hyphae and fruiting bodies (×100 magnification, Safranin-O); (**J**) hyphae and fruiting bodies (×100 magnification, PAS stain); (**K**) hyphae and fruiting bodies (×100 magnification, Alcian Blue stain); (**L**) hyphae and spores (×100 magnification, GMS stain).

**Figure 2 microorganisms-10-01226-f002:**
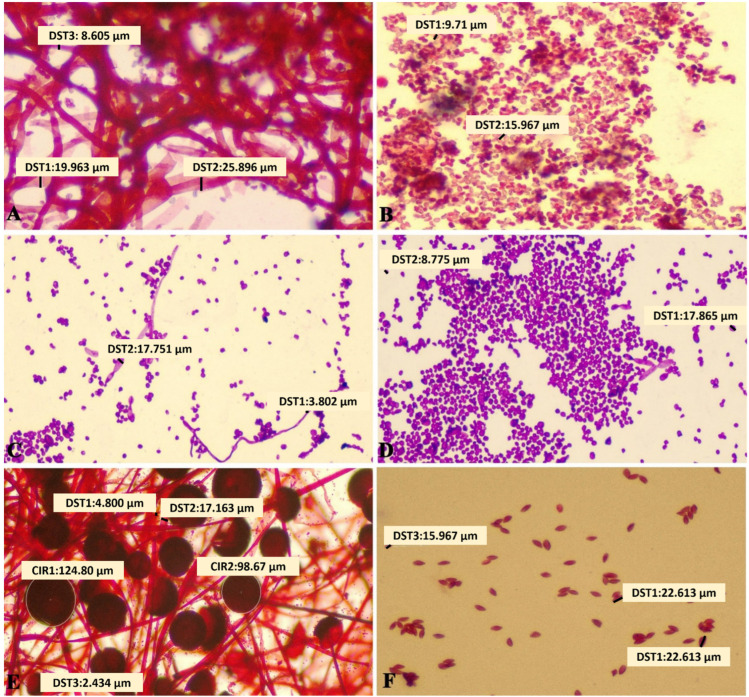
Magnification: ×100 magnification, Safranin–O Stain. Morphometric analysis exhibiting measurements obtained using Gryphax image analysis software. *Aspergillus Flavus*: (**A**) hyphae; (**B**) spores. *Candida albicans*: (**C**) hyphae and spores; (**D**) spores and germ tube formation. *Rhizopus Oryzae*: (**E**) hyphae and fruiting bodies; (**F**) spores. (White calibrations represent measurement).

**Table 1 microorganisms-10-01226-t001:** Comparison of staining intensity of spores of *Candida albicans*, *Aspergillus flavus* and *Rhizopus oryzae*.

Staining Intensity			Scoring			Pearson Chi-Square	*p*-Value
			1	2	3		
** *Candida albicans* **	Safranin-O	Count	0	2	8	40.914	**0.0001**
		% within Staining Intensity	0.0%	20.0%	80.0%		
	Alcian Blue	Count	8	2	0		
		% within Staining Intensity	80.0%	20.0%	0.0%		
	PAS	Count	0	8	2		
		% within Staining Intensity	0.0%	80.0%	20.0%		
	GMS	Count	8	2	0		
		% within Staining Intensity	80.0%	20.0%	0.0%		
** *Aspergillus flavus* **	Safranin-O	Count	0	1	9	22.947	**0.001**
		% within Staining Intensity	0.0%	10.0%	90.0%		
	Alcian Blue	Count	1	8	1		
		% within Staining Intensity	10.0%	80.0%	10.0%		
	PAS	Count	1	8	1		
		% within Staining Intensity	10.0%	80.0%	10.0%		
	GMS	Count	0	2	8		
		% within Staining Intensity	0.0%	20.0%	80.0%		
** *Rhizopus oryzae* **	Safranin-O	Count	1	8	1	49.253	**0.0001**
		% within Staining Intensity	10.0%	80.0%	10.0%		
	Alcian Blue	Count	9	1	0		
		% within Staining Intensity	90.0%	10.0%	0.0%		
	PAS	Count	1	8	1		
		% within Staining Intensity	10.0%	80.0%	10.0%		
	GMS	Count	0	1	9		
		% within Staining Intensity	0.0%	10.0%	90.0%		

**Table 2 microorganisms-10-01226-t002:** Comparison of staining intensity of hyphae of *Candida albicans*, *Aspergillus flavus* and *Rhizopus oryzae*.

Staining Intensity			Scoring			Pearson Chi-Square	*p*-Value
			1	2	3		
** *Candida albicans* **	Safranin-O	Count	0	1	9	58.400	**0.0001**
		% within Staining Intensity	0.0%	10.0%	90.0%		
	Alcian Blue	Count	1	9	0		
		% within Staining Intensity	10.0%	90.0%	0.0%		
	PAS	Count	0	9	1		
		% within Staining Intensity	0.0%	90.0%	10.0%		
	GMS	Count	9	1	0		
		% within Staining Intensity	90.0%	10.0%	0.0%		
** *Aspergillus flavus* **	Safranin-O	Count	0	2	8	26.444	**0.0001**
		% within Staining Intensity	0.0%	20.0%	80.0%		
	Alcian Blue	Count	1	8	1		
		% within Staining Intensity	10.0%	80.0%	10.0%		
	PAS	Count	1	9	0		
		% within Staining Intensity	10.0%	90.0%	0.0%		
	GMS	Count	0	1	9		
		% within Staining Intensity	0.0%	10.0%	90.0%		
** *Rhizopus oryzae* **	Safranin-O	Count	0	1	9	55.614	**0.0001**
		% within Staining Intensity	0.0%	10.0%	90.0%		
	Alcian Blue	Count	0	8	2		
		% within Staining Intensity	0.0%	80.0%	20.0%		
	PAS	Count	0	9	1		
		% within Staining Intensity	0.0%	90.0%	10.0%		
	GMS	Count	9	1	0		
		% within Staining Intensity	90.0%	10.0%	0.0%		

**Table 3 microorganisms-10-01226-t003:** Comparison of staining intensity of fruiting bodies (yeast/conidia/sporangia) of *Candida albicans*, *Aspergillus flavus* and *Rhizopus oryzae*.

Staining Intensity			Scoring			Pearson Chi-Square	*p*-Value
			1	2	3		
** *Candida albicans* **	Safranin-O	Count	0	1	9	59.354	**0.0001**
		% within Staining Intensity	0.0%	10.0%	90.0%		
	Alcian Blue	Count	1	8	1		
		% within Staining Intensity	10.0%	80.0%	10.0%		
	PAS	Count	0	9	1		
		% within Staining Intensity	0.0%	90.0%	10.0%		
	GMS	Count	10	0	0		
		% within Staining Intensity	100.0%	0.0%	0.0%		
** *Aspergillus flavus* **	Safranin-O	Count	0	0	10	40.000	**0.0001**
		% within Staining Intensity	0.0%	0.0%	100.0%		
	Alcian Blue	Count	0	0	10		
		% within Staining Intensity	0.0%	0.0%	100.0%		
	PAS	Count	0	10	0		
		% within Staining Intensity	0.0%	100.0%	0.0%		
	GMS	Count	0	0	10		
		% within Staining Intensity	0.0%	0.0%	100.0%		
** *Rhizopus oryzae* **	Safranin-O	Count	0	1	9	56.800	**0.0001**
		% within Staining Intensity	0.0%	10.0%	90.0%		
	Alcian Blue	Count	0	1	9		
		% within Staining Intensity	0.0%	10.0%	90.0%		
	PAS	Count	1	9	0		
		% within Staining Intensity	10.0%	90.0%	0.0%		
	GMS	Count	9	1	0		
		% within Staining Intensity	90.0%	10.0%	0.0%		

**Table 4 microorganisms-10-01226-t004:** Comparison of mean morphologic differentiation (spore size) of *Candida albicans*, *Aspergillus flavus* and *Rhizopus oryzae*.

Morphologic Differentiation (Morph Diff)			Scoring			χ^2^	*p*-Value
			1	2	3		
** *Candida albicans* **	Safranin-O	Count	0	0	10	80.00	**0.001**
		% within Morph Diff	0.0%	0.0%	100.0%		
	Alcian Blue	Count	0	10	0		
		% within Morph Diff	0.0%	100.0%	0.0%		
	PAS	Count	0	0	10		
		% within Morph Diff	0.0%	0.0%	100.0%		
	GMS	Count	10	0	0		
		% within Morph Diff	100.0%	0.0%	0.0%		
** *Aspergillus flavus* **	Safranin-O	Count	0	0	10	40.00	**0.001**
		% within Morph Diff	0.0%	0.0%	100.0%		
	Alcian Blue	Count	0	10	0		
		% within Morph Diff	0.0%	100.0%	0.0%		
	PAS	Count	0	0	10		
		% within Morph Diff	0.0%	0.0%	100.0%		
	GMS	Count	0	0	10		
		% within Morph Diff	0.0%	0.0%	100.0%		
** *Rhizopus oryzae* **	Safranin-O	Count	0	0	10	40.00	**0.001**
		% within Morph Diff	0.0%	0.0%	100.0%		
	Alcian Blue	Count	0	10	0		
		% within Morph Diff	0.0%	100.0%	0.0%		
	PAS	Count	0	0	10		
		% within Morph Diff	0.0%	0.0%	100.0%		
	GMS	Count	0	0	10		
		% within Morph Diff	0.0%	0.0%	100.0%		

Chi-squared test, *p*-value < 0.05—statistically significant.

**Table 5 microorganisms-10-01226-t005:** Comparison of mean morphologic differentiation (spore shape) of *Candida albicans*, *Aspergillus flavus* and *Rhizopus oryzae*.

Morphologic Differentiation (Morph Diff)			Scoring			χ^2^	*p*-Value
			1	2	3		
** *Candida albicans* **	Safranin-O	Count	0	0	10	80.00	**0.001**
		% within Morph Diff	0.0%	0.0%	100.0%		
	Alcian Blue	Count	0	10	0		
		% within Morph Diff	0.0%	100.0%	0.0%		
	PAS	Count	0	0	10		
		% within Morph Diff	0.0%	0.0%	100.0%		
	GMS	Count	10	0	0		
		% within Morph Diff	100.0%	0.0%	0.0%		
** *Aspergillus flavus* **	Safranin-O	Count	0	0	10	80.00	**0.001**
		% within Morph Diff	0.0%	0.0%	100.0%		
	Alcian Blue	Count	0	10	0		
		% within Morph Diff	0.0%	100.0%	0.0%		
	PAS	Count	0	0	10		
		% within Morph Diff	0.0%	0.0%	100.0%		
	GMS	Count	10	0	0		
		% within Morph Diff	100.0%	0.0%	0.0%		
** *Rhizopus oryzae* **	Safranin-O	Count	0	0	10	80.00	**0.001**
		% within Morph Diff	0.0%	0.0%	100.0%		
	Alcian Blue	Count	0	10	0		
		% within Morph Diff	0.0%	100.0%	0.0%		
	PAS	Count	0	0	10		
		% within Morph Diff	0.0%	0.0%	100.0%		
	GMS	Count	10	0	0		
		% within Morph Diff	100.0%	0.0%	0.0%		

Chi-squared test, *p*-value < 0.05—statistically significant.

**Table 6 microorganisms-10-01226-t006:** Comparison of mean morphologic differentiation (nature of hyphae—pseudohyphae/septate/non-septate) of *Candida albicans*, *Aspergillus flavus* and *Rhizopus oryzae*.

Morphologic Differentiation (Morph Diff)			Scoring			χ^2^	*p*-Value
			1	2	3		
** *Candida albicans* **	Safranin-O	Count	0	0	10	82.00	**0.001**
		% within Morph Diff	0.0%	0.0%	100.0%		
	Alcian Blue	Count	10	0	0		
		% within Morph Diff	100.0%	0.0%	0.0%		
	PAS	Count	0	10	0		
		% within Morph Diff	0.0%	100.0%	0.0%		
	GMS	Count	10	0	0		
		% within Morph Diff	100.0%	0.0%	0.0%		
** *Aspergillus flavus* **	Safranin-O	Count	0	0	10	82.00	**0.001**
		% within Morph Diff	0.0%	0.0%	100.0%		
	Alcian Blue	Count	0	10	0		
		% within Morph Diff	0.0%	100.0%	0.0%		
	PAS	Count	0	10	0		
		% within Morph Diff	0.0%	100.0%	0.0%		
	GMS	Count	10	0	0		
		% within Morph Diff	100.0%	0.0%	0.0%		
** *Rhizopus oryzae* **	Safranin-O	Count	0	0	10	41.00	**0.001**
		% within Morph Diff	0.0%	0.0%	100.0%		
	Alcian Blue	Count	10	0	0		
		% within Morph Diff	100.0%	0.0%	0.0%		
	PAS	Count	0	0	10		
		% within Morph Diff	0.0%	0.0%	100.0%		
	GMS	Count	10	0	0		
		% within Morph Diff	100.0%	0.0%	0.0%		

Chi-squared test, *p*-value < 0.05—statistically significant.

**Table 7 microorganisms-10-01226-t007:** Comparison of mean morphologic differentiation (nature of branching—angulation/germ tube formation) of *Candida albicans*, *Aspergillus flavus* and *Rhizopus oryzae*.

Morphologic Differentiation (Morph Diff)			Scoring			χ^2^	*p*-Value
			1	2	3		
** *Candida albicans* **	Safranin-O	Count	0	0	10	82.00	**0.001**
		% within Morph Diff	0.0%	0.0%	100.0%		
	Alcian Blue	Count	10	0	0		
		% within Morph Diff	100.0%	0.0%	0.0%		
	PAS	Count	0	10	0		
		% within Morph Diff	0.0%	100.0%	0.0%		
	GMS	Count	10	0	0		
		% within Morph Diff	100.0%	0.0%	0.0%		
** *Aspergillus flavus* **	Safranin-O	Count	0	0	10	82.00	**0.001**
		% within Morph Diff	0.0%	0.0%	100.0%		
	Alcian Blue	Count	0	10	0		
		% within Morph Diff	0.0%	100.0%	0.0%		
	PAS	Count	0	10	0		
		% within Morph Diff	0.0%	100.0%	0.0%		
	GMS	Count	10	0	0		
		% within Morph Diff	100.0%	0.0%	0.0%		
** *Rhizopus oryzae* **	Safranin-O	Count	0	0	10	41.00	**0.001**
		% within Morph Diff	0.0%	0.0%	100.0%		
	Alcian Blue	Count	10	0	0		
		% within Morph Diff	100.0%	0.0%	0.0%		
	PAS	Count	0	0	10		
		% within Morph Diff	0.0%	0.0%	100.0%		
	GMS	Count	10	0	0		
		% within Morph Diff	100.0%	0.0%	0.0%		

Chi-squared test, *p*-value < 0.05—statistically significant.

**Table 8 microorganisms-10-01226-t008:** Morphometric analysis of mean size of the spores/hyphae/fruiting body of 50 microscopic fields.

Organism	Spores (μm)		Hyphae (μm)		Fruiting Bodies Diameter (μm)		*p* Value
	**Smallest**	**Largest**	**Smallest**	**Largest**	**Smallest**	**Largest**	**0.001**
*Aspergillus* *flavus*	9.71	15.96	**8.60**	**25.85**	18.94	33.46	
*Candida* *albicans*	8.77	17.86	3.80	17.75	-	-	
*Rhizopus* *oryzae*	**15.96**	**28.25**	2.43	17.16	**98.67**	**124.80**	

Chi-squared test, *p*-value < 0.05—statistically significant.

## Data Availability

Not applicable.

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
