# Peer review of "Alternate Special Stains for the Detection of Mycotic Organisms in Oral Cyto-Smears—A Histomorphometric Study"

_microorganisms, 2022, doi:10.3390/microorganisms10061226_

Round 1

Reviewer 1 Report

The work is very well presented and written. But, I would like to ask the authors two questions: 1/ When carrying out certain stains, it is advisable to carry out a control to check that the stain has been carried out correctly. As in the study, they already work with the microorganisms to study, these same organisms would act as controls. But in a sample where it is unknown whether the microorganism is present or not, the authors consider that it would be convenient to control the staining. Could you explain the answer, whether it is affirmative or negative? 2/ the stains have a component of subjectivity due to this fact the stains should have been observed by two different people to see if the observations of the stains were similar between both people.

Author Response

Dear Reviewer, 

We thank you for the comments

Regards

Authors

Reviewer 2 Report

The authors present a histo-morphometric study using alternate special stains for the detection of mycotic organisms in oral cyto-smears. The title accurately reflects the article. The study involves an important area of health and presents a clear and clinically useful message. The manuscript is well written in terms of clarity, style, and use of English and has a logical construction. The discussion section explains the case in the context of published information. The conclusions accurately and clearly explain the main clinical message. The figures are of good quality and relevant to the clinical message. The references are appropriate and current.

Overall, the manuscript is of high quality and presents a novel histochemical marker (Safranin-O) for the detection of mycotic organisms. The manuscript is highly recommended for publication.

Author Response

(The authors gave the same response as above.)
